# Changes in Nutritional Status in Pulmonary Tuberculosis: Longitudinal Changes in BMI According to Acid-Fast Bacilli Smear Positivity

**DOI:** 10.3390/jcm9124082

**Published:** 2020-12-17

**Authors:** Yousang Ko, Changwhan Kim, Yong Bum Park, Eun-Kyung Mo, Jin-Wook Moon

**Affiliations:** 1Department of Internal Medicine, Division of Pulmonary, Allergy and Critical Care Medicine, Kangdong Sacred Heart Hospital, Hallym University College of Medicine, Seoul 05355, Korea; bfspark2@gmail.com (Y.B.P.); ekmopark@gmail.com (E.-K.M.); cozybar@hanmail.net (J.-W.M.); 2College of Medicine, Lung Research Institute of Hallym University, Chuncheon 24253, Korea; 3Department of Internal Medicine, School of Medicine, Jeju National University Hospital, Jeju National University, Jeju 63241, Korea; masque70@naver.com

**Keywords:** pulmonary tuberculosis, body mass index, smear positivity, malnutrition

## Abstract

Malnutrition is closely associated with pulmonary tuberculosis (PTB). However, changes before and after treatment remain unclear. We aimed to investigate the longitudinal changes in nutritional status from treatment to follow-up of TB in 215 PTB cases in South Korea. First, we evaluated the trend in body mass index (BMI) from the time of diagnosis to a 2-year follow-up. Second, we compared the BMIs of our cases with 5694 controls who participated in a Korean national survey after treatment. During the treatment period, the BMI of the smear-positive group (*n* = 72) significantly increased compared with that of the smear-negative group (*n* = 143) (+1.9 kg/m^2^ vs. +0.4 kg/m^2^, *p* = 0.001). Almost all the changes occurred in the early phase, with unremarkable differences in the rest of the treatment period and up to the 2-year follow-up period. When compared with controls, the smear-positive PTB group also had a lower BMI than the smear-negative PTB group, which, however, was lower than that of the general population, though all the participants regained their BMIs during treatment. These results clarify the nutritional aspects of PTB and enable better strategies to support patients with PTB.

## 1. Introduction

Tuberculosis (TB), caused by *Mycobacterium tuberculosis* (MTB), is the most lethal infectious disease in the world and has afflicted humans for centuries. Pulmonary TB (PTB) is the most common TB infection in humans [1]. The association between the development of active TB and malnutrition has been described on the basis of geographical correlation, animal models, and observational studies in humans [2,3,4,5].

The mechanism by which malnutrition leads to PTB occurrence is not yet clear. Malnutrition may reduce the proliferation of T lymphocytes and impair cell-mediated immunity, which in turn may lead to increased susceptibility to infection [2]. In contrast, obesity is considered a protective factor against active PTB [6,7]. Moreover, malnutrition is an important factor in the development of PTB and is also considered to be associated with the relapse risk and treatment response rate of PTB [8,9].

Efforts have been made to terminate TB infections under the banner of the WHO End TB strategy under the Stop TB plan. The WHO predicted that malnutrition is responsible for 18% of the global TB burden, attributable for 1.9 million cases. Moreover, it is the most powerful risk factor as compared to human immunodeficiency virus (HIV) infection, smoking, diabetes, and alcohol consumption [1]. Thus, a better understanding of malnutrition-related TB infections will help us reach the goal we desire.

Malnutrition status can be measured using diverse methods. Among these, body mass index (BMI) and mid-upper arm circumference are commonly used to assess nutritional status [2]. In adults, BMI has been extensively used in clinical practice despite its limitations, as it has the advantage of easy accessibility. Previous studies have shown correlations between BMI and TB infections [2,10,11].

Previous research has focused on body weight and its changes in patients with PTB [8,12,13,14]. Body weight loss is often observed in patients with tuberculosis at the time of diagnosis and, for centuries, has been considered to be an important risk factor for tuberculosis. On the contrary, body weight gain is a useful biomarker of treatment outcome and relapse [8]. However, there are problems of inter-personal variability and inter-racial differences. Previous studies have investigated the effect of anti-TB treatment on body weight gain, but they were mostly based on patients with smear-positive PTB. However, smear-negative PTB is more common in developed countries, including South Korea.

In the present study, we hypothesized that the changes in malnutrition status differ according to disease severity, because of “nothing to lose nothing to gain”. We aimed to evaluate longitudinal changes in BMI in patients with PTB with positive and negative sputum acid-fast bacilli (AFB) smears at the time of diagnosis, on treatment completion, and up to 2 years after treatment completion. Moreover, we compared changes in serologic markers with changes in BMI.

## 2. Experimental Section

### 2.1. Study Population and Design

This study was a retrospective analysis of protocol-based collective data form the TB clinic of Hallym University Kangdong Sacred Heart Hospital (Seoul, Korea) from January 2015 to December 2016. The hospital is situated in an area with intermediate TB burden, with a reported estimated prevalence of 101/100,000 persons in 2015. We enrolled all consecutive patients aged >20 years who had been diagnosed with culture-positive PTB. All the patients with PTB received anti-TB medication daily. Patients were excluded if they were transferred to another hospital, if they died or were lost to follow-up during anti-TB treatment, if they showed poor adherence to anti-TB treatment, or if they had incomplete medical records. We also excluded cases with multi-drug or rifampicin-resistant PTB because of the long duration of anti-TB treatment.

Another group was recruited for comparison. We aimed to determine whether the nutritional status of patients with PTB could be completely recovered to match that of the general population. Thus, we compared the nutritional status of the enrolled patients at the time of TB treatment completion with that of the subjects who participated in the Korea National Health and Nutrition Examination Surveys (KNHANES) in 2013. KNHANES is a nationwide survey of the health and nutritional status of the Korean population (https://knhanes.cdc.go.kr/knhanes). We evaluated BMI after treatment completion to compare it with normal subjects in the general population. Finally, 5694 subjects who participated in the survey were selected after excluding those with previous (*n* = 223) and current (*n* = 4) TB infections.

The protocol for this study was approved by the Institutional Review Board of Hallym University Kangdong Sacred Heart Hospital (IRB 2020-04-008). Informed consent was waived because of the retrospective nature of the study.

### 2.2. Data Collection

We reviewed the medical records of the patients at our TB clinics. Finally, 215 adult patients with PTB were included in this study (Figure 1).

### 2.3. Radiographic Data

Chest radiographs and computerized tomographic scans of the patients were reviewed for the presence of cavities and to determine the extent of the disease. The extent of each lesion was categorized based on lobar involvement. Uni-lobar and multi-lobar involvement of PTB were defined by the involvement of ≤1 and ≥2 lobes, respectively.

### 2.4. Microbiological Data

The AFB smears were examined after auramine-rhodamine fluorescent staining and were graded on a scale from 0 to 4+ [15]. MTB culturing was simultaneously performed with solid media, 3% Ogawa media (Eiken Chemical, Tokyo, Japan) and liquid media in the mycobacteria growth indicator tube 960 system (BD Biosciences, Franklin Lakes, NJ, USA).

### 2.5. Assessment of BMI

BMI was used to assess nutritional status, calculated as the weight in kilograms divided by the height in meters squared. Individuals were categorized into five BMI groups: severely underweight (<16), underweight (16–18.4), normal weight (18.5–24.9), overweight (25–29.9), and obese (≥30) [2]. Malnutrition is defined as BMI less than 18.5 kg/m^2^ [16].

### 2.6. Treatment Outcomes of PTB

Treatment outcomes were categorized based on WHO definitions as follows: (1) cured PTB patients who had bacteriologically confirmed TB at the beginning of treatment and who were smear- or culture-negative in the last month of treatment and on at least one previous occasion; (2) patients with TB who completed the treatment (without evidence of failure, but with no record of being cured); (3) patients whose treatments failed (patients with TB whose sputum smears or cultures were positive at month 5 or later during the treatment); (4) patients who died (patients with TB who died due to any reason before or during the course of treatment) [17].

### 2.7. Statistical Analysis

Data are presented as means and standard deviations (SD) for continuous variables and numbers (percentage) for categorical variables. Data were compared using Student’s t-test for continuous variables and Pearson’s chi-square test or Fisher’s exact test for categorical variables. Longitudinal changes in BMI were analyzed by mixed-effect linear regressions. All the tests were two-sided and *p*-values < 0.05 were considered to be statistically significant. Data were analyzed using IBM SPSS Statistics, version 26 (IBM, Armonk, NY, USA) and Graph Pad Prism 8.0 (GraphPad, San Diego, CA, USA).

## 3. Results

### 3.1. Patient Characteristics

During the 2-year-long study period, 227 patients were treated for PTB, and subsequently 12 were excluded in accordance with the exclusion criteria (Figure 1). A total of 215 patients with PTB, consisting of 72 (33.5%) AFB smear-positive cases and 143 (66.5%) AFB smear-negative cases were included for analysis. The demographic and clinical characteristics of the enrolled cases are summarized in Table 1. The mean age of the population was 50.9 ± 18.9 years, of which 136 (63.3%) were men. The mean body weight and BMI were 58.3 ± 11.4 kg and 21.2 ± 3.2 kg/m^2^, respectively. The patients with PTB were concomitantly affected by TB pleuritis in 34 cases (15.8%), TB lymphadenitis, 4 cases (1.9%); and, TB epididymitis, 1 case (0.4%).

### 3.2. Clinical, Radiological, and Microbiological Differences between the AFB Smear-Positive and AFB Smear-Negative Groups

There were unremarkable differences in age, height, body weight, comorbidities, and smoking status according to AFB smear status before anti-TB treatment at baseline. However, the proportions of male patients (72.2% vs. 58.7, *p* = 0.036) and heavy alcohol drinkers (12.5 vs. 7.7%, *p* < 0.001) were higher in the AFB smear-positive group (Table 1).

Although all the enrolled patients predominantly had localized PTB, the AFB smear-positive group had more extensive PTB than the smear-negative group (63.9 vs. 39.0%, *p* < 0.001) (Table 2). The frequency of cavitation was also considerably higher in the smear-positive group (61.2 vs. 18.1%, *p* < 0.001). In the patients with drug-resistant PTB, there were unremarkable differences between the two groups except for the proportion of patients resistant to pyrazinamide.

In terms of nutrition status, the AFB smear-positive group had a lower BMI (20.6 ± 3.3 kg/m^2^ vs. 21.5 ± 3.1 kg/m^2^, *p* = 0.033) than the AFB smear-negative group. Moreover, in the severely underweight, underweight, normal weight, overweight, and obese categories of BMI, the AFB smear-positive group had more undernourished patients (*p* = 0.006) (Table 1). Biochemically, the AFB smear-positive group had a lower albumin level than the smear-negative group (3.7 ± 0.6 g/dL vs. 4.0 ± 0.6 g/dL, *p* < 0.001) at the time of diagnosis (Table 3).

### 3.3. Sequential Changes in Body Weight, BMI, Serum Protein Levels, and Albumin Levels during Anti-TB Therapy

The mean changes in body weight, body weight percentages compared with baseline, BMIs, serum protein levels, and serum albumin levels from initiation to completion of anti-TB therapy are presented in Figure 2 and Table 3. The curves of observed changes from baseline in the BMI, and in serum albumin levels in the two groups, separated early but did not continue to diverge after the first trimester. In other words, the AFB smear-positive group had more malnutrition in the baseline compared with the negative group, but these differences were not observed after the first trimester. After first trimester, mean values of body weight and BMI in the AFB smear-positive group were higher, but not statistically different. This indicates that catch-up recovery of undernutrition in AFB smear-positive PTB occurred in the early period of anti-TB therapy. During the treatment period, the patients with AFB smear-positive PTB had significantly higher regains of BMI than the patients with AFB smear-negative PTB (+1.9 kg/m^2^ vs. +0.4 kg/m^2^, *p* = 0.001).

### 3.4. Sequential Changes in Body Weight and BMI after Anti-TB Therapy

The changes in body weight, percentages of body weight change compared with the baseline, and BMIs from enrolment to 2 years after anti-TB therapy are presented in Figure 3. The trend observed after treatment completion did not differ between the two groups. Furthermore, there were unremarkable changes in body weight and BMI.

Additionally, we compared the BMIs and nutritional statuses, according to the BMI categories, of patients at treatment completion and at the restoration point from TB infection with subjects who participated in the KNHANES in 2013 (*n* = 5694) (Table 4). Despite the successful completion of treatment for TB, the PTB group had a lower BMI and the patients were more undernourished than the general population.

## 4. Discussion

We investigated the longitudinal changes in BMI as a marker of malnutrition in patients with PTB. The factors that may affect malnutrition were analyzed by clinical, radiological, and laboratory examination according to the microbiological burden. Our study showed that undernutrition was more prevalent in patients with positive AFB smears and in those who had more advanced disease, and that patients almost recovered from TB infection-related undernutrition in the early phase after initiation of anti-TB treatment. This is remarkable in patients with AFB smear-positive, and not in AFB smear-negative patients. However, nutritional status did not reach that of the general population despite the completion of anti-TB treatment. Moreover, this phenomenon was maintained for a follow-up period of 2 years.

TB and malnutrition have been traditionally considered to be strongly associated [11]. However, it is unclear whether malnutrition leads to TB infection or whether it results from TB infection. Moreover, one previous study showed that this phenomenon is only observed in PTB and not in extra-PTB [18]. This could be affected by the characteristics of extra-PTB such as its occurrence in relatively young and pauci-bacillary infections. This means that the correlation between TB infections and malnutrition may be more complex than we think and expect [2,11]. Therefore, long-term observations in the general population need to be performed to thoroughly investigate changes in nutritional statuses before and after TB infection. However, this poses a challenge in real-world settings.

In our study, the nutritional status of TB patients, as measured by BMI, improved in the early phase but did not change in at least two years in the AFB smear-positive group. This indirectly suggests that malnutrition may predispose patients to PTB infection. However, there were unremarkable changes in the AFB smear-negative group. They had no remarkable BMI recovery during or after anti-TB treatment. This means that there was no remarkable malnutrition in the AFB smear-negative group. It also raises the assumption that PTB infection can cause malnutrition due to it being a consumptive disease. Furthermore, the BMI after 2 years of treatment completion was relatively lower in the AFB smear-negative group than that in the general population. The present finding that malnutrition was not fully repleted during anti-TB treatment is consistent with the results of a previous study in a small group of 30 subjects in the United Kingdom [19]. In addition, in our study, recovery was not satisfactory after treatment. Therefore, it indicates that there is insufficient understanding about the relationship between TB infections and nutrition. Thus, a well-designed prospective study is needed for better understanding.

TB infection remains a major health problem worldwide, including in South Korea. Fortunately, in South Korea, it has steadily declined to 46.4 per 100,000 population, which was the estimated incidence in 2019 [20]. South Korea is constantly moving toward being a low-TB burden country. However, the TB burden is still not as low as that in other OECD (Organization for Economic Co-operation and Development) countries. TB-related mortality is also continuously decreasing in South Korea [20]; however, PTB survivors face other malnutrition-related medical problems in the future even after the end of treatment [21]. Thus, it is necessary to better understand not only anti-TB treatment but also host-related responses, including nutritional changes. A comprehensive understanding of TB and TB-related phenomena could lead us one step forward towards the real end of TB in the world.

There were several limitations to this study. First, given its retrospective nature, selection bias may have influenced our findings. For example, we included and analyzed cases from TB outpatient clinics because they could measure BMI and they conducted follow-ups regularly. This means that severe cases with prolonged hospitalization due to advanced PTB or other combined diseases such as malignancies, bedridden states, or deaths during treatment were not included because the trend of malnutrition could not be identified in them. Therefore, this could underestimate our results. However, advanced cases of PTB that may lead to fatality, or complex cases with other medical problems, are also important to overcome TB infection, but these are not a major proportion. Second, this study was conducted in a high-income country with a very low HIV infection rate. Thus, this limitation could also underestimate our results. Third, our results were based on a relatively small sample size. For these reasons, our results may not be generalized to other clinical situations, and further studies are needed to determine whether these results can be applied.

The current study provides new data on the association between PTB and nutrition changes. Malnutrition-related TB is usually found in advanced disease, and is recovered in the early stages of treatment. However, the recovery of nutritional state is not sufficiently comparable to the general population after the end of treatment ended and after 2 years of follow-up. Therefore, the completion of anti-TB treatment might be the beginning, but not the end, from a social health perspective.

## Figures and Tables

**Figure 1 jcm-09-04082-f001:**
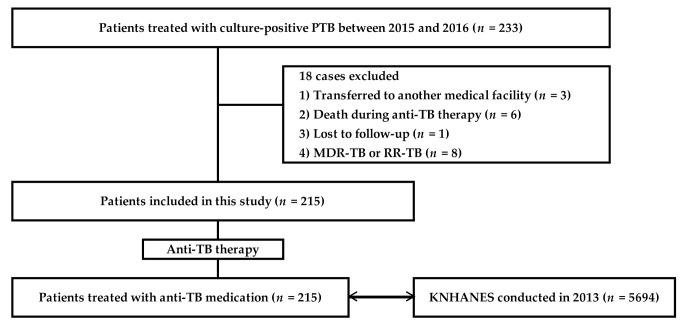
Flow chart of the study. PTB, pulmonary tuberculosis; MDR, multi-drug resistant; TB, tuberculosis; RR, rifampicin resistance; KNHANES, Korea National Health and Nutrition Examination Survey.

**Figure 2 jcm-09-04082-f002:**
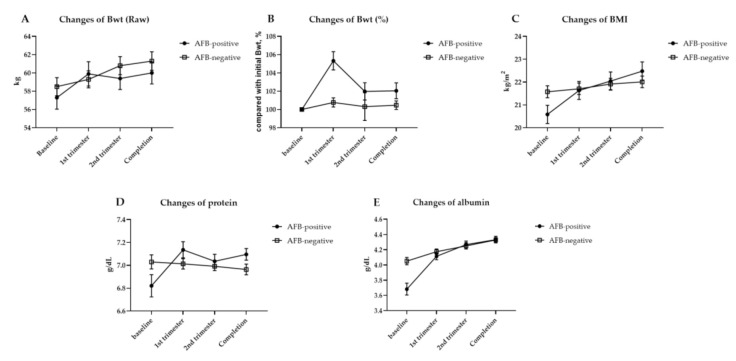
Changes from baseline in body weight (**A**), % changes in body weight compared with baseline (**B**), body mass index (**C**), serum protein levels (**D**), and serum albumin levels (**E**) during anti-TB treatment; the observed mean changes from baseline over the period of anti-TB therapy in the AFB smear-positive and negative groups. Bwt, body weight; BMI, body mass index; AFB, acid-fast bacilli.

**Figure 3 jcm-09-04082-f003:**
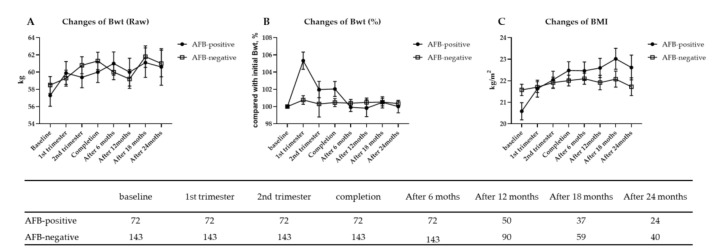
Changes from baseline in body weight (**A**), % change of body weight compared with baseline (**B**) body mass index (**C**) from the time of diagnosis to 2 years follow-up after treatment completion; the observed mean change from baseline over the period of anti-TB therapy in the AFB smear-positive and negative groups. Bwt, body weight; BMI, body mass index; AFB, acid-fast bacilli.

**Table 1 jcm-09-04082-t001:** Demographic and clinical characteristics of the enrolled 215 patients with PTB.

	All Patients	AFB-S (+) PTB	AFB-S (−) PTB	*p*-Value
	*n* = 215	*n* = 72	*n* = 143	
Age, years	50.9 ± 18.9	54.2 ± 17.1	49.3 ± 19.6	0.075
Male gender, %	136 (63.3)	52 (72.2)	84 (58.7)	0.036
Height, cm	165.3 ± 9.5	166.2 ± 8.8	164.9 ± 9.8	0.347
Body Weight, kg	58.0 ± 11.4	57.9 ± 11.3	58.7 ± 11.4	0.643
BMI, kg/m^2^	21.2 ± 3.2	20.6 ± 3.3	21.5 ± 3.1	0.033
severely underweight (<16)	6 (2.8)	3 (4.2)	3 (2.1)	0.006
underweight (16–18.4)	38 (17.7)	22 (30.6)	16 (11.2)	
normal weight (18.5–24.9)	144 (67.0)	41 (56.9)	103 (72.0)	
overweight (25–29.9)	25 (11.6)	6 (8.3)	19 (13.3)	
obese (>30)	2 (0.9)	0	2 (1.4)	
Comorbidity				
COPD or asthma	9 (3.8)	1 (1.4)	8 (5.6)	0.278
Thyroid disease	2 (0.9)	0	2 (1.4)	0.552
Cardiovascular disease	32 (13.7)	12 (16.7)	20 (14.0)	0.685
Malignancy	11 (4.7)	5 (6.9)	6 (4.2)	0.513
Hematologic disease	2 (0.9)	2 (2.8)	0	0.111
Chronic liver disease	19 (8.1)	8 (11.1)	11 (7.7)	0.449
Rheumatic disease	6 (2.6)	1 (1.4)	5 (3.5)	0.666
CKD	2 (0.9)	1 (1.4)	1 (0.7)	0.559
Diabetes	44 (14.8)	22 (30.6)	22 (15.4)	0.012
Neurologic disease	6 (2.6)	0	6 (4.2)	0.182
Cerebrovascular disease	2 (0.9)	1 (1.4)	1 (0.7)	1.000
Immunologic disease	1 (0.4)	0	1 (0.7)	1.000
Silicosis	1 (0.4)	0	1 (0.7)	1.000
KTP	2 (0.9)	1 (1.4)	1 (0.7)	1.000
Smoking status				0.223
Never smoker	92 (39.3)	26 (36.1)	66 (46.2)	
Former smoker	47 (20.1)	15 (20.8)	32 (22.4)	
Current smoker	76 (32.5)	31 (43.1)	45 (31.5)	
Alcohol use				<0.001
Heavy drinker		9 (12.5)	11 (7.7)	

AFB-S, acid-fast bacilli smear; PTB, pulmonary tuberculosis; BMI, body mass index; CKD, chronic kidney disease; KTP, kidney transplantation, COPD, chronic obstructive lung disease.

**Table 2 jcm-09-04082-t002:** Clinical, radiological and microbiological characteristics of the enrolled 215 cases of PTB.

	All Patients	AFB-S (+) PTB	AFB-S (−) PTB	*p*-Value
	*n* = 215	*n* = 72	*n* = 143	
New cases	187 (87)	61 (84.7)	126 (88.1)	0.523
Previously treated cases	28 (13.0)	11 (15.3)	17 (11.9)	
Radiographic feature				
Cavity	64 (29.8)	41 (61.2)	23 (18.1)	<0.001
Extent of lung lesion				
Uni-lobar involvement	116 (54.0)	26 (36.1)	91 (70.0)	<0.001
Multi-lobar involvement	85 (39.5)	46 (63.9)	39 (30.0)	
AFB smear status				
Positive	72 (33.5)			
4+	16 (7.4)	16 (22.2)		
3+	19 (8.8)	19 (26.4)		
2+	18 (8.4)	18 (25.0)		
1+	19 (8.8)	19 (26.4)		
Negative	143 (66.5)			
DST profiles				
Resistant to isoniazid	20 (9.3)	10 (13.9)	10 (7.0)	0.134
Resistant to pyrazinamide	7 (3.3)	5 (6.9)	2 (1.4)	0.043
Resistant to ethambutol	2 (0.9)	2 (2.8)	0	0.111
Resistant to streptomycin	5 (2.3)	1 (1.4)	4 (2.8)	0.666
Resistant to fluoroquinolone	2 (0.9)	1 (1.4)	1 (0.7)	1.000
Resistant to protionamide	3 (1.4)	1 (1.4)	2 (1.4)	1.000
Resistant to PAS	4 (1.9)	3 (4.2)	1 (0.7)	0.110

AFB-S, acid-fast bacilli smear; PTB, pulmonary tuberculosis; DST, drug sensitivity test; PAS, para-amino salicylic acid.

**Table 3 jcm-09-04082-t003:** Sequential changes of the nutritional status in enrolled 215 cases after anti-TB treatment.

	All Patients	AFB-S (+) PTB	AFB-S (−) PTB	*p*-Value
	*n* = 215	*n* = 72	*n* = 143	
Treatment duration, months	8.3 ± 3.4	9.2 ± 3.8	7.2 ± 1.2	0.003
BMI, baseline	21.2 ± 3.2	20.6 ± 3.3	21.5 ± 3.1	0.033
Severely underweight (<16)	6 (2.8)	3 (4.2)	3 (2.1)	0.006
Underweight (16–18.4)	38 (17.7)	22 (30.6)	16 (11.2)	
Normal weight (18.5–24.9)	144 (67.0)	41 (56.9)	103 (72.0)	
Overweight (25–29.9)	25 (11.6)	6 (8.3)	19 (13.3)	
Obese (>30)	2 (0.9)	0	2 (1.4)	
BMI, 1st trimester	21.6 ± 3.1	21.6 ± 3.4	21.6 ± 2.9	0.737
BMI, 2nd trimester	21.9 ± 3.1	22.1 ± 3.4	21.9 ± 2.9	0.722
BMI, completion	22.1 ± 3.1	22.5 ± 3.4	21.9 ± 2.9	0.254
Severely underweight (<16)	3 (1.4)	1 (1.4)	2 (1.4)	0.316
Underweight (16–18.4)	16 (7.4)	6 (8.3)	10 (7.0)	
Normal weight (18.5–24.9)	155 (75.5)	47 (65.3)	108 (75.7)	
Overweight (25–29.9)	37 (17.2)	15 (20.8)	22 (15.4)	
Obese (>30)	4 (1.9)	3 (4.2)	1 (0.7)	
Protein, baseline	6.9 ± 0.8	6.8 ± 0.8	7.0 ± 0.7	0.108
Protein, 1st trimester	7.1 ± 0.6	7.2 ± 0.6	7.0 ± 0.5	0.137
Protein, 2nd trimester	7.0 ± 0.5	7.0 ± 0.5	6.9 ± 0.5	0.47
Protein, completion	7.0 ± 0.5	7.1 ± 0.4	7.0 ± 0.6	0.086
Albumin, baseline	3.9 ± 0.6	3.7 ± 0.6	4.0 ± 0.6	<0.001
Albumin, 1st trimester	4.2 ± 0.4	4.1 ± 0.3	4.1 ± 0.4	0.323
Albumin, 2nd trimester	4.2 ± 0.5	4.2 ± 0.3	4.2 ± 0.5	0.995
Albumin, completion	4.3 ± 0.4	4.3 ± 0.3	4.3 ± 0.4	0.648

AFB-S, acid-fast bacilli smear; BMI, body mass index.

**Table 4 jcm-09-04082-t004:** Comparison of demographics between enrolled 215 cases on completion of anti-TB treatment and 5694 cases of KHANES data in 2013.

	In This Study	KNHANES 2013	*p*-Value
	*n* = 215	*n* = 5694	
Age, years	51.5 ± 18.9	50.3 ± 16.3	0.345
Male gender, %	136 (63.3)	2240 (42.9)	<0.001
Height, cm	165.3 ± 9.5	162.3 ± 9.4	<0.001
Body Weight, kg	60.9 ± 11.5	62.8 ± 11.9	0.015
BMI, completion, kg/m2	22.1 ± 3.1	23.7 ± 3.5	<0.001
severely underweight (<16)	3 (1.4)	10 (0.2)	<0.001
underweight (16–18.4)	16 (7.4)	232 (4.1)	
normal weight (18.5–24.9)	155 (75.5)	3295 (57.9)	
overweight (25–29.9)	37 (17.2)	1853 (32.5)	
obese (>30)	4 (1.9)	304 (5.3)	

KNHANES, Korea National Health and Nutrition Examination Surveys; BMI, body mass index.

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
