# Peer review of "Changes in Nutritional Status in Pulmonary Tuberculosis: Longitudinal Changes in BMI According to Acid-Fast Bacilli Smear Positivity"

_jcm, 2020, doi:10.3390/jcm9124082_

Round 1

Reviewer 1 Report

The study design is not adequate. The study group runs from January 2015 to December 2016 and consists of 215 patients. This group cannot be compared with a group of 3,200 cases collected in a longer period of time from 2015 to 2018. Both groups could be compared if the period of time were the same from January 2015 to December 2016, and in this second group would have collected the same variables as the first group of patients to compare if there could be differences between the two groups as the origin of the population is different. Because this is not the case and only the baseline characteristics of the patients in the second group at the time of diagnosis have been collected, I do not believe that this second group contributes anything to the study. On the contrary, I think it produces more confusion. For this reason, I believe that this second group of 3,200 cases of tuberculosis patients should not be included in the study.

Author Response

Response to the Reviewers

We are submitting our point-by-point responses to the reviews and have revised our manuscript according to Reviwer`s comments. We have carefully considered each of the comments and criticisms offered by the reviewers. Specific changes are marked in highlight in this revised manuscript.

First Revision of jcm-1032948

# Response to Reviewer 1’s Comments

General comments

The study design is not adequate. The study group runs from January 2015 to December 2016 and consists of 215 patients. This group cannot be compared with a group of 3,200 cases collected in a longer period of time from 2015 to 2018. Both groups could be compared if the period of time were the same from January 2015 to December 2016, and in this second group would have collected the same variables as the first group of patients to compare if there could be differences between the two groups as the origin of the population is different. Because this is not the case and only the baseline characteristics of the patients in the second group at the time of diagnosis have been collected, I do not believe that this second group contributes anything to the study. On the contrary, I think it produces more confusion. For this reason, I believe that this second group of 3,200 cases of tuberculosis patients should not be included in the study.

Response. We understand the reviewer’s concern and agree. We excluded 3,200 cases of PTB patients in our study. We remake figure 1 and delete table 4. We revised the information related with second group of 3,200 cases in Abstract, Method, Results and Discussion sections. (Line 18-19, 76-77, 133-134 and 249-251)

Reviewer 2 Report

Comments to the authors:

The authors aimed to evaluate longitudinal changes in body mass index (BMI) in patients with PTB with positive and negative sputum acid-fast bacilli smears at the time of diagnosis, on treatment completion, and up to 2 years after treatment completion.

This was an appreciable effort to understand changes in BMI before and after anti-TB treatment.

Minor revisions

I would just like to suggest some changes that can improve the article and make it easier to read:

1) Replace "malnutrition" by "nutritional" in article title (Line 2). Taking into account that 67.0% of the enrolled cases of PTB had normal weight (18.5–24.9) before anti-TB treatment.

2) Term "malnutrition" is used throughout the text, but its definition is missing. It was only mentioned body mass index (BMI) groups: severely underweight (<16), underweight (16–18.4), normal weight (18.5–24.9), overweight (25–29.9) and obese (≥30) (Lines 109-110).

3) To try to replace by the most suitable reference for "Pulmonary TB (PTB) is the most common TB infection in humans" (Lines 32-33).

4) The titles of tables must be more explanatory. I suggest including the number of cases for "enrolled cases", "other cohort of PTB" and "KHANES data of 2013” (Lines 143, 168, 171,173, 208-209).

5) To check carefully same data that was used in 2 different tables:

Table 1. Demographic and clinical characteristics of the enrolled patients with PTB (Line 143).

BMI, kg/m2 21.3 ± 3.2   20.9 ± 3.4   21.5 ± 3.1   0.225

and Table 3. Sequential changes in the nutritional status after anti-TB treatment (Line 171).

BMI, baseline 21.2 ± 3.2   20.6 ± 3.3   21.5 ± 3.1   0.033

6) To correct "treatmen.t" in the table title (Table 3, line 171).

7) To develop a little in the discussion what are "better strategies to support patients with PTB" (Lines 26-27). Nutritional support?

This study aimed to identify factors that influence adherence to established therapy in people with pulmonary tuberculosis, in Portugal, and to analyse the relationship between knowledge about tuberculosis and its treatment and the age of the participants.

The study was conducted in Pneumological Diagnosis Centers, in a sample of 303 people with pulmonary tuberculosis, in the North of Portugal.

To answer the research questions, quantitative and qualitative methods were used. However, it is not clear how adherence to treatment and associated factors were assessed.

The authors are not comfortable with statistical analysis.

This study has weak internal and external validity.

Additional note:

The reference 020/2019 of the favourable opinion of the Ethics Committee for Health of the Northern Region Health Administration, which indicated by authors, corresponds to the "Projeto de intervenção na parentalidade positiva em relação ao medo desenvolvimental nas crianças/adolescentes".

Author Response

Response to the Reviewers

We are submitting our point-by-point responses to the reviews and have revised our manuscript according to Reviwer`s comments. We have carefully considered each of the comments and criticisms offered by the reviewers. Specific changes are marked in highlight in this revised manuscript.

First Revision of jcm-1032948

# Response to Reviewer 2’s Comments

General comments

The authors aimed to evaluate longitudinal changes in body mass index (BMI) in patients with PTB with positive and negative sputum acid-fast bacilli smears at the time of diagnosis, on treatment completion, and up to 2 years after treatment completion.

This was an appreciable effort to understand changes in BMI before and after anti-TB treatment.

I would just like to suggest some changes that can improve the article and make it easier to read:
Response. We appreciate the reviewer’s words of encouragement and helpful comments that have substantially improved the quality of our paper. We are submitting a revised manuscript that addresses the concerns raised. A detailed, point-by-point response to these concerns is attached.

Specific comments

C1. Replace "malnutrition" by "nutritional" in article title (Line 2). Taking into account that 67.0% of the enrolled cases of PTB had normal weight (18.5–24.9) before anti-TB treatment.

R1. Thank you for the suggestions. We have modified the article title of the revised manuscript as you recommended (Line 2)

C2. Term "malnutrition" is used throughout the text, but its definition is missing. It was only mentioned body mass index (BMI) groups: severely underweight (<16), underweight (16–18.4), normal weight (18.5–24.9), overweight (25–29.9) and obese (≥30) (Lines 109-110).

R2. Thank you for your comment. We have modified the related information in the Method section of the revised manuscript. (Line 105)

C3. To try to replace by the most suitable reference for "Pulmonary TB (PTB) is the most common TB infection in humans" (Lines 32-33).

R3. We appreciate the reviewer’s comments. We asked the check-up of references format to our medical library before submission. We think there's an error during that process. We apologize for our mistake once more. We revised the reference number 1. (Line 31-32)

C4. The titles of tables must be more explanatory. I suggest including the number of cases for "enrolled cases", "other cohort of PTB" and "KHANES data of 2013” (Lines 143, 168, 171,173, 208-209).

R4. Thank you for your comment. We revised the titles of tables as you recommended. (Line 137, 157, 160 and 195-196)

C5. To check carefully same data that was used in 2 different tables:

Table 1. Demographic and clinical characteristics of the enrolled patients with PTB (Line 143).

BMI, kg/m2 21.3 ± 3.2   20.9 ± 3.4   21.5 ± 3.1   0.225

and Table 3. Sequential changes in the nutritional status after anti-TB treatment (Line 171).

BMI, baseline 21.2 ± 3.2   20.6 ± 3.3   21.5 ± 3.1   0.033

R5. We appreciate the reviewer’s helpful comment and apologize for our big mistake. We revised the Table 1. (Line 137)

C6. To correct "treatmen.t" in the table title (Table 3, line 171).

R6. We appreciate the reviewer’s comments. We apologize our careless. We revised the sentence in Table 3. (Line 160)

C7-1. To develop a little in the discussion what are "better strategies to support patients with PTB" (Lines 26-27). Nutritional support?

R7-1. Thank you for your comment. We think that nutritional support and effort may be better to focus on early phase of anti-TB treatment based on our finding. In addition, TB patients were still remained to be more undernutrition despite successfully treated. It could help to make a plan of supporting TB cases under and after treatment.

C7-2.

This study aimed to identify factors that influence adherence to established therapy in people with pulmonary tuberculosis, in Portugal, and to analyse the relationship between knowledge about tuberculosis and its treatment and the age of the participants.

The study was conducted in Pneumological Diagnosis Centers, in a sample of 303 people with pulmonary tuberculosis, in the North of Portugal.

To answer the research questions, quantitative and qualitative methods were used. However, it is not clear how adherence to treatment and associated factors were assessed.

R7-2. We are afraid that we are not quite clear what you mean by above comments. We aimed to evaluate the changes of BMI for long-time period and performed this study using 227 cases of TB clinic in South Korea. We’d appreciate it if you could explain in a little more detail about meaning of your comment.

C7-3. The authors are not comfortable with statistical analysis.

 R7-3. We understand the reviewer’s concern and agree. So, we analyzed our data with help from clinical epidemiologist worked in Hallym Research Institute of Clinical Epidemiology, Chuncheon, Republic of Korea.

C7-4. This study has weak internal and external validity.

R7-4. We understand the reviewer’s concern and agree. Thus, we tried to overcome this weakness by comparing it with a large population as much as possible.

C8. Additional note: The reference 020/2019 of the favourable opinion of the Ethics Committee for Health of the Northern Region Health Administration, which indicated by authors, corresponds to the "Projeto de intervenção na parentalidade positiva em relação ao medo desenvolvimental nas crianças/adolescentes".

R8. We are not quite clear what you mean by the comments. We are sorry but could you explain in a little more detail once more?